# *Clostridioides difficile* Infection in Chronic Kidney Disease—An Overview for Clinicians

**DOI:** 10.3390/jcm10020196

**Published:** 2021-01-07

**Authors:** Sylwia Dudzicz, Andrzej Wiecek, Marcin Adamczak

**Affiliations:** Department of Nephrology, Transplantation and Internal Medicine, Medical University of Silesia, 40-027 Katowice, Poland; sdudzicz@sum.edu.pl (S.D.); madamczak@sum.edu.pl (M.A.)

**Keywords:** *Clostridioides difficile* infection, chronic kidney disease, dysbiosis, probiotic

## Abstract

Increased incidence of *Clostridioides difficile* infection (CDI), occurrence of severe and complicated CDI, and more frequent occurrence of drug-resistant, recurrent or non-hospital CDI has become a worldwide clinical problem. CDI is more common in patients with chronic kidney disease (CKD) than in the general population. CDI seems to be associated with frequent hospitalization, frequently used antibiotic therapy, dysbiosis, and abnormalities of the immune system observed in CKD patients. Dysbiosis is a common disorder found in CKD patients. It may be related to insufficient fiber content in the diet, reduced amount of consumed fluids and often reduced physical activity, constipation, impaired gastrointestinal motility, multidrug pharmacotherapy, and uremic milieu in CKD stage 5. In patients with CKD the clinical manifestations of CDI are similar to the general population; however, more frequent recurrence of CDI and higher prevalence of severe CDI are reported. Moreover, the increase in CDI related mortality is observed more in CKD patients than in the general population. The aim of this review paper is to summarize the current knowledge concerning the epidemiology, pathogenesis, clinical picture, and prevention and treatment in CKD patients.

## 1. *Clostridioides difficile* Infection

*Clostridioides difficile* is an anaerobic gram-positive bacterium with the ability to produce spores. In 2016, based on genetic analysis, it was reclassified from the genus *Clostridium* to the genus *Clostridioides*. *Clostridioides difficile* causes diarrhea associated with the use of antibacterial drugs. *Clostridioides difficile* infection (CDI) was considered primarily as a nosocomial infection but, in recent years, *Clostridioides difficile* more and more often causes diarrhea in non-hospitalized patients also. Exposure to antibacterial agents is a major CDI risk factor [1].

## 2. Epidemiology of *Clostridioides difficile* Infection

In recent decades, an increased incidence of CDI, occurrence of severe and complicated CDI, and more frequent occurrence of drug-resistant, recurrent or non-hospital CDI has been observed. All the above mentioned changes in CDI epidemiology might be related to the worldwide spread of the hypervirulent endemic strain *Clostridioides difficile* BI/NAP1/027 [2]. The increased virulence of the BI/NAP1/027 strain is associated with increased production of toxins A and B, production of binary toxin, greater ability to form spores and more frequent resistance to fluoroquinolones. In patients infected with the BI/NAP1/027 *Clostridioides difficile* strain, severe and complicated forms of CDI are more frequent. Relapses and higher mortality (three times higher compared to strains 001 and 014) are also observed [3]. In a study completed in Poland by Pituch et al. (part of the *European Clostridium difficile Infections Surveillance Network* (ECDIS-Net)) it was found that as many as 62% of all identified *Clostridioides difficile* strains in 13 hospitals in Poland included in this study belonged to this hypervirulent strain [4]. However, based on the result of epidemiological study analyzing the incidence of CDI in 2011–2017 in the United States, in recent years a tendency to reduction in the incidence of the health care–associated strain was observed. At the same time, with stabilization at a high level, the incidence of community-acquired CDI was noted [5].

CDI is more common in patients with chronic kidney disease (CKD) than in the general population. It seems to be associated with more frequent hospitalization, more frequently used antibiotic therapy, usually multidrug pharmacotherapy, dysbiosis and abnormalities of the immune system observed in CKD patients. These immune system deficiencies might be due to over-used immunosuppressive therapy or to uremic toxicity which occurs in patients with advanced CKD.

Keddis et al., based on data obtained from *The National Hospital Discharge Survey* (NHDS) including data from 162 million hospitalizations in the years 2005–2009 in the United States, found an almost two-fold higher incidence of CDI in patients with CKD compared to patients without CKD (1.5% vs. 0.7%) [6]. Moreover, dialysis CKD patients were more likely to suffer from CDI than non-dialysis CKD patients (odds ratio (OR), 1.33; 95% CI, 1.32–1.35; *p* < 0.001). Based on NHDS data, it can also be concluded that the incidence of CDI in CKD patients increases with the advance of CKD. In the studied population, CDI was most common in the patients with CKD stage 5 during dialysis therapy (44% of study group). The CDI frequency in the group of patients with CKD stage 3, 4 or 5 was 22% and in the group of patients with CKD stage 1 or 2 it was the lowest among all groups (2%) (Table 1) [6]. A meta-analysis of 20 epidemiological studies completed by Phatharacharukul et al. indicates a significantly increased risk of CDI and its recurrence in patients with CKD. In addition, the degree of increased risk of CDI in patients treated with dialysis was higher than in patients with less advanced CKD. In comparison to the general population, there is an increased risk of CDI in the group of patients with CKD treated with dialysis and the increased risk of recurrence of CDI in CKD patients was found (Table 1) [7].

Another meta-analysis of 19 studies by Thongprayoon et al. showed that patients with CKD are characterized by an increased risk of severe CDI, recurrent CDI and an increased risk of CDI related mortality (Table 1) [8].

In a study of 207 hospitalized CKD patients with CDI, 67% of patients were in CKD stage 5, whereas only 6% were in CKD stage 1 [9]. Kim et al. also observed 513 patients with CKD and demonstrated that the incidence of CDI in the dialysis patients was higher than in patients with CKD stage 4 or 5 (OR, 3.34 vs. 2.90, respectively) [10]. There was also a higher in-hospital mortality due to CDI in patients with more advanced CKD (CKD stages 4 and 5) [10].

## 3. Pathogenesis of *Clostridioides difficile* Infection

CDI spreads via the fecal-oral route and is caused by the ingestion of spores which are resistant to high temperatures, acids and antibacterial drugs. After ingestion, spores under the influence of contact with bile acids in the duodenal lumen transform into vegetative forms of bacteria. Under favorable conditions, such as antibiotic-induced dysbiosis, the vegetative form of *Clostridioides difficile* may colonize the large intestine. In the large intestine, *Clostridioides difficile* produces enterotoxins and transforms the vegetative forms of this bacteria into spores, in the process of so-called sporulation [1,11]. The main pathogenic factor of *Clostridioides difficile* is the production of bacterial toxins: A and B which inactivate Rho GTP-ase leading to depolymerization of actin fibers and to damage to the colon epithelial cells. The presence of these toxins in the intestinal lumen also stimulates production of tumor necrosis factor alpha (TNF α) and pro-inflammatory interleukins (mainly IL-1β and IL-18), as well as, the production of oxygen free radicals. Moreover, it facilitates the recruitment of neutrophils and macrophages, increases vascular permeability, opens epithelial junctions and stimulates apoptosis of epithelial cells. Damage to the intestinal mucosa and excessive mucus production results in diarrhea, colitis, and the formation of pseudo-membranes consisting of inflammatory cells, fibrous exudates and necrosis [12,13].

## 4. Risk Factors for *Clostridioides difficile* Infection

The most important CDI risk factor is exposure to antibacterial agents, especially those with a broad spectrum of antibacterial activity (penicillin, cephalosporins, fluoroquinolones and clindamycin). These drugs (especially fluoroquinolones, penicillin and cephalosporins) are often used in CKD patients (and others in urinary tract infection therapy). Other CDI risk factors are older age, recent hospitalization, accompanying comorbidity (especially cancer), CKD, liver cirrhosis, diabetes, inflammatory bowel disease, disorders of the immune system (patients infected with human immunodeficiency virus (HIV), patients after solid organ transplant or hematopoietic stem cells), post-gastrointestinal surgery or endoscopy, and the use of drugs that inhibit the production of gastric acid, such as proton pump inhibitors or type 2 histamine receptor antagonists [14,15].

The main CDI risk factors in CKD patients are frequent infections, use of wide-spectrum antibiotic therapy, more frequent hospitalizations than in the general population, hypoalbuminemia and impaired immune system function [16]. According to the Berman et al. analysis, the incidence of infections in dialysis patients was 5.7 events per 1000 dialysis-days. The most common infections in these patients are central venous catheter infections, lower limb skin and soft tissues infections and pneumonia [17]. More frequent infections and frequent infections with multidrug-resistant bacterial strains also result in the use of broad-spectrum antibiotic therapy, which increases CDI risk. More frequent hospitalizations will also increase the exposure to contact with an asymptomatic *Clostridioides difficile* carrier or with a symptomatic CDI patient. There are no published data on the impact of specific comorbidities on CDI incidence in CKD patients and especially in patients undergoing renal replacement therapy. Considering that the above-mentioned nephrological conditions are CDI risk factors, it seems reasonable to assume that additional chronic co-morbidities such as cancer, inflammatory bowel disease or impaired function of the immune system may increase the probability of CDI occurrence.

## 5. The Role of Dysbiosis in CDI Pathogenesis

The human gut microbiota consists of microorganisms that live in the human gastrointestinal tract. The gene pool of all microorganisms in a microbiota is referred to as the microbiome. The intestinal microbiota consists of numerous species of bacteria, viruses, fungi and protozoa [18]. The colon has the highest density of bacterial colonization and species diversity. In this part of the digestive tract, over 400 species of bacteria can be identified, such as anaerobic bacteria from the *Bacteroides*, *Bifidobacteria*, *Fusobacteria* and *Pepto-streptococci* groups and aerobic and relatively anaerobic bacteria from the *Enterobacteria* and *Lactobacillus* groups [19]. Intact human gut microbiota fulfills many functions important for the host. It positively influences the differentiation and growth of the intestinal epithelium and maintains the integrity of the intestinal barrier, provides energetic substances for intestinal epithelial cells through the production of short-chain fatty acids in the process of decomposition of food residues, and produces B vitamins and vitamin K. Short-chain fatty acids also affect the intestinal motility, modulate the inflammatory response, regulate glucose metabolism, ensure the proper functioning of the intestinal barrier by stimulating the production of proteins that are part of tight junctions, and reduce intestinal pathogenic microorganisms colonization by stimulation of the intestinal epithelium to produce mucin and antimicrobial peptides (intestinal bacteriocins) and by reduction the pH of the intestinal content [20,21]. Intestinal bacteria, due to their species diversity and the ability to produce many different enzymes not produced by human cells, participate in the digestion and fermentation of certain nutrients (e.g., fiber, fructo-oligosaccharides, oligosaccharides), obtaining easily digestible metabolites. The gut microbiota also plays an important immunomodulatory role: it stimulates the proper development of the immune system, regulating its activity, e.g., reducing excessive inflammatory reaction.

Dysbiosis is an imbalance in the composition and functions of microbiota. It is defined as a decrease in the number of beneficial bacteria, an increase in the number of potentially pathogenic bacteria, and a loss of microbiota diversity. Dysbiosis may be associated with various diseases of the digestive, nervous, respiratory and circulatory systems.

Dysbiosis is a common disorder in CKD patients. In these patients, an increase in the number of bacteria such as *Firmicutes*, *Actinobacteria* and *Proteobacteria* and a decrease in the number of *Bifidobacteria* and *Lactobacillus* were found [22]. There are several explanations for the causes of dysbiosis in CKD patients. Due to insufficient fiber content in the diet, a limited amount of consumed fluids and often limited physical activity, constipation and impaired gastrointestinal motility are more frequently observed in these patients, which may lead to increased growth of bacteria in the intestinal lumen [23,24]. In addition, the above-mentioned low supply of dietary fiber also leads to a reduction in the number of *Bifidobacteria*, and thus a reduction in the production of short-chain fatty acids. This can lead to increased inflammation in the gut and unsealing of the intestinal barrier. Due to the increased concentration of urea and uric acid in the plasma, these substances are also excreted in greater amounts into the lumen of the gastrointestinal tract, and their presence creates environmental conditions favoring the growth of the bacteria producing urease, uricase and enzymes, metabolizing phenolic and indole compounds, among others, to ammonia, which also damages the intestinal barrier. Damage to the intestinal barrier causes excessive absorption of uremic toxins from the gastrointestinal tract and the possibility of transfer of pathogenic microorganisms beyond the gastrointestinal tract into the blood. This may lead to chronic inflammation and as a consequence to damage of the vascular endothelium, participating in the pathogenesis of atherosclerosis and cardiovascular disease [25,26]. In interventional studies completed on CKD patients and hemodialysis patients, during which probiotic preparations were administered, a decrease in the concentration of proinflammatory cytokines, urea and uric acid in the blood plasma, improvement of the digestive system function and improvement of the quality of life assessed using the quality of life (QOL) scale were found [27,28]. In addition, the results of a meta-analysis of 11 clinical trials by Chung et al. suggest a relationship between the early stages of deterioration in kidney function and the increasing number of bacteria producing uremic toxin [29].

## 6. The Role of Probiotics and Prebiotics in Intestinal Microbiota Modification

Modifications of the composition of the intestinal microbiota can be obtained, among other ways, through the use of probiotic preparations. Probiotics are live microorganisms whose supply brings health benefits to the host organism. Desirable features of microorganisms that allow their use in probiotic preparations are lack of pathogenic or toxic properties, antagonistic activity against pathogenic bacteria of the gastrointestinal tract (e.g., production of antibacterial substances), active growth and ability to colonize the large intestine, genetic stability, and lack of genes associated with antibiotic resistance. Probiotics can be microorganisms found in the natural healthy microflora of the human colon—the most commonly used strains are *Lactobacillus*, *Bifidobacterium*, *Streptococcus* and *Sacharomyces*. It is also important to stress that the probiotic preparations are administered in the right amount—they should contain from 10^9^ to 10^10^ colony forming units (CFUs) of live microorganisms. There are single-strain and multi-strain preparations available and their individual types differ in their biological properties and, therefore, their indications for use. Risks related to the use of probiotics may include bacteremia or fungemia (case reports) and transfer of antibiotic resistance genes to other bacteria (in the case of incorrect selection of the probiotic strain by the manufacturer) [30,31].

Prebiotics are non-digestible chemicals that, when ingested, stimulate the growth and activity of bacteria living in the large intestine, such as *Bifidobacterium* and *Lactobacillus*, and therefore benefit the health of the host. Prebiotics include polysaccharide A, peptidoglycans, inulin and oligosaccharides. Syn-biotics are preparations that combine probiotic strains with prebiotic substances [30,31].

## 7. Clinical Presentation of *Clostridioides difficile* Infection

The clinical presentation depends both on the toxicity of the *Clostridioides difficile* strain and on the general condition of the patient. Exposure to toxigenic *Clostridioides difficile* strains may result as asymptomatic colonization of the large intestine or lead to symptomatic CDI. Persistent intestinal colonization by *Clostridioides difficile* is found in 4–15% of the population, and periodic colonization has been demonstrated in most humans. Those patients who develop IgG antibodies to toxins A and B during exposure to *Clostridioides difficile* do not have CDI symptoms. In addition, immunity acquired during the first CDI (especially when IgG antibodies against toxin A are present) protects against disease recurrence [11,32]. Clinical symptoms of CDI mainly involve the gastrointestinal tract. Depending on the severity of the infection, they can be very different. CDI may present as diarrhea of varying severity without signs of inflammatory bowel disease, colitis without pseudo-membranous membranes, pseudomembranous colitis, fulminant colitis complicated by acute distension of the colon (*megacolon toxicum*), or paralytic ileus or colonic perforation [33].

The most common clinical symptoms of CDI are watery diarrhea, fever, nausea, vomiting, lower abdominal pain and malaise [34,35]. They usually begin two or three days after exposure to *Clostridioides difficile*. In 90% of cases, patients with CDI received antimicrobial therapy one to eight weeks prior to the onset of symptoms. In case of CDI due to antimicrobial therapy, clinical symptoms usually appear between the fifth and tenth day of use of the antibacterial agents. In 20% of patients with CDI, the infection recurs after treatment. Recurrent CDI is defined as a relapse of the infection till eight weeks after the previous episode. The risk factors for recurrent CDI are age over 65 years, advanced underlying disease and antibacterial treatment after treatment of an earlier episode of CDI [36]. The causes of CDI recurrence include disturbances in colonization of the large intestine by the physiological bacterial flora, the presence of vegetative forms in the gastrointestinal tract, less frequent non-production of IgG antibodies against toxins A and B, or resistance of *Clostridioides difficile* to the antibacterial drugs used, i.e., vancomycin or fidaxomicin [37].

In patients with CKD the clinical manifestations of CDI are similar to those of the general population [16]. However, more frequent recurrences of CDI and higher prevalence of severe CDI are reported. In the study by Kujawa-Szewieczek et al. in which over 85% of the subjects were patients with CKD, diarrhea with various severity, abdominal pain, vomiting and fever were main CDI symptoms, occurring in 100%, 59%, 11% and 78% patients, respectively [38]. In a retrospective case-control study by Abdelfatah et al., recurrence of CDI was found in 10% of CDI patients. The most frequent relapses occurred in patients with multiple comorbidities including CKD (OR, 1.3; 95% CI, 1.0–2.4; *p* = 0.039) [39]. More frequent CDI relapses in CKD may be caused by more frequent hospitalizations, more frequent use of broad-spectrum antibiotics, and abnormalities of the immune system function in these patients.

A specific group of patients treated with renal replacement therapy are those treated with the peritoneal dialysis technique. These patients, in addition to the common risk factors in patients undergoing renal replacement therapy, have also an increased risk of developing CDI peritonitis. This is caused by a bacterial translocation from the intestinal lumen into the peritoneal cavity. CDI can also complicate the course of peritoneal dialysis-related peritonitis, because cephalosporins are the main antibiotics used in the treatment of such peritonitis. Their use is associated with an increased risk of developing CDI. The initial symptoms of peritonitis associated with the peritoneal dialysis technique may be clinically indistinguishable from CDI. In both cases, peritoneal dialysis-related peritonitis complicated by CDI and peritonitis caused by bacterial translocation, CDI patients will complain of abdominal pain, tenderness, nausea, vomiting and diarrhea, and the dialysate will be cloudy. The differentiation between these two conditions during the first days of infection depends mainly on the stool test for *Clostridioides difficile* toxins, as well as on the clinical resolution of the peritonitis symptoms after initiation of treatment dedicated to peritoneal dialysis-related peritonitis [40,41].

## 8. Diagnosis of *Clostridioides difficile* Infection

CDI diagnostic procedures in CKD patients are the same as in subjects from the general population. The current guidelines for the diagnosis of CDI were published by the *Infectious Diseases Society of America* (IDSA) and the *Society for Healthcare Epidemiology of America* (SHEA) in 2017. CDI is diagnosed by the presence of diarrhea or *megacolon toxicum* and one of the following criteria: presence of toxins A and/or B in the stool sample or presence of toxin producing strain *Clostridioides difficile* in stool culture, or identification of pseudomembranous enteritis on endoscopy or during surgery or if pseudomembranous enteritis is recognized by histopathology. The material for laboratory tests is a stool sample taken from a patient with suspected CDI. In the absence of diarrhea in the intestinal obstruction, rectal swabs should be obtained for molecular testing or cultures to detect the presence of the toxin-causing strain of *Clostridioides difficile*. The material collected for testing should be submitted to the laboratory within 2 h of collection. If this is not possible, the material should be stored at 4 °C for no longer than 72 h, because toxins A and B are thermolabile [11,42].

In the diagnosis of CDI, several tests differing in sensitivity, specificity and methodology are used. Low sensitivity and moderate specificity tests are enzyme immunoassays detecting toxins A and B. High sensitivity and low or moderate specificity tests are nucleic acid amplification tests (NAAT), enzyme-linked immunosorbent assay glutamate dehydrogenase (EIA GDH), or culturing the strain and determining its toxigenic culture (TC). NAAT molecular tests detect genes encoding *Clostridioides difficile* toxins. It should be remembered that they may also be positive in asymptomatic carriers of the toxigenic strain of *Clostridioides difficile*. The GDH antigen test detects both toxigenic and non-toxigenic strains of *Clostridioides difficile* and may cross-react with other *Clostridioides species*. Due to their high sensitivity, the NAAT and EIA GDH tests are often used in the first stage of the CDI diagnostic algorithm. The stool culture for *Clostridioides difficile* and the determination of toxigenicity allows the detection of vegetative and spore forms [11].

Currently, a multi-step diagnostic algorithm is recommended. In the first stage, it is recommended to perform a high-sensitivity test—NAAT or EIA GDH. In case of a positive result, in the second stage enzyme immunoassays are performed to detect toxins A and B. A positive result of this test confirms the presence of CDI. In the case of a negative result, it is advisable to identify CDI or the carrier of a toxigenic strain of *Clostridioides difficile* based on the patient’s clinical evaluation or by performing the NAAT test (in the case where we performed the GDH EIA in the first stage) or performing stool culture for *Clostridioides difficile* and determining its toxinogenicity [11,42].

## 9. Prophylaxis of *Clostridioides difficile* Infection

CDI prevention should consist of increasing the level of hygiene and eliminating risk factors as much as possible. After CDI diagnosis, the patient should be isolated in a room with bathroom and toilet, and isolation may be completed 48 h after the end of infection symptoms. An important element of CDI prophylaxis is frequent hand washing with soap, which allows the spores to be mechanically rinsed off the skin surface before and after each contact with the patient and his surroundings. It is not recommended to disinfect hands with alcohol-based preparations, as their effectiveness is lower than appropriate hand washing. During contact with the patient, medical personnel should wear disposable personal protective equipment, such as gloves and aprons. Appropriate disinfection of surfaces (agents containing hypochlorite), medical equipment and items used by the patient is also recommended. Patients are advised to use the shower to regularly remove spores from the skin surface and lower the seat before flushing the toilet to prevent aerosol production with suspended spores. In order to reduce the risk factors, it is recommended to use antibacterial drugs only when there are relevant indications (the increased CDI risk persists for three months after the end of antibacterial treatment). If it is necessary to start antibacterial treatment, it is recommended to choose, if possible, an antibacterial drug with a lower CDI risk (carbapenems, macrolides, aminoglycosides, tigecycline, tetracyclines, trimethoprim with sulfamethoxazole). The use of non-steroidal anti-inflammatory drugs, proton pump inhibitors or histamine 2 receptor antagonists should be limited only to clinical situations in which there are absolute indications for treatment with these drugs and only for a strictly defined period of time [14,15,43].

Among the preventive measures against CDI, the use of probiotics should also be discussed. The normal microbiota of the host’s gut prevents the conversion of spores to vegetative forms of *Clostridioides difficile* and prevents their growth. The administration of probiotics during antimicrobial treatment aims to prevent changes in the composition of the intestinal flora, which is one of the causes of CDI. The results of studies on the use of probiotics in the prevention of antibiotic diarrhea associated with *Clostridioides difficile* infection are inconclusive [44]. However, in some studies it has been shown in the general population that probiotics are effective in preventing CDI in patients undergoing antibacterial treatment [45,46].

In the largest randomized clinical trial conducted for assessing the use of probiotics as a prophylaxis of CDI (number of patients—2941, PLACIDE study), it was not shown that a preparation containing four bacterial strains: *Lactobacillus acidophilus* CUL60, *Lactobacillus acidophilus* CUL21, *Bifidobacterium bifidum* CUL20 and *Bifidobacterium lactis,* reduces CDI risk [44]. In a meta-analysis of 18 studies involving data from 6851 participants, Johnston et al. found that probiotics reduced the risk of CDI by 66% [45]. Also, in a meta-analysis of 39 randomized trials of 9955 patients, Goldenberg et al. LAO showed that the administration of probiotics was associated with a lower risk of CDI compared with placebo or without prophylaxis during antimicrobial treatment (1.5% vs. 4.0%, *p* < 0.001) [46].

In the case of using probiotics for CDI prevention, there are still many unknowns, such as which probiotic, in what dose and how long should it be used to obtain a beneficial effect. The mechanisms of action of probiotics that may prove beneficial in CDI prevention are intestinal mucosa colonization, mucin production stimulation by intestinal mucosa cells, antibacterial substances production, immune system locally stimulation, and ability to compete with pathogenic microorganisms for nutrients and living place.

One of the probiotic strains of bacteria that has found application in CDI prevention is *Lactobacillus plantarum 299v* (LP299v). It is a gram-positive lactobacillus, without antibiotic resistance genes, which has the ability to adhere to intestinal mucosa cells in vitro, stimulates mucin production by intestinal mucosa cells, and produces antibacterial substances such as plantacin A and plantaricin S/k83 [47,48,49]. Klarin et al. in a study of 44 patients hospitalized in the intensive care unit found no colonization of *Clostridioides difficile* in patients receiving LP299v compared to patients who did not use this probiotic (0% vs. 19%; *p* < 0.05) [50]. In a double-blind, placebo-controlled clinical trial by Lönnermark et al. reduction in the severity of gastrointestinal side effects, such as nausea and diarrhea, in patients undergoing antimicrobial therapy who used LP299v was found [42].

The efficacy of CDI prophylaxis with LP299v was also assessed in the population of CKD patients. In a retrospective study, Dudzicz et al. analyzed the frequency of infections caused by *Clostridioides difficile* in hospitalized patients in the nephrology ward before, during and after the administration of LP299v administration as a routine prophylaxis of CDI in patients during antibacterial and immunosuppressive treatment. This strain was administered once daily in the form of oral capsules containing 10 × 10^9^ CFU LP299v. After the implementation of the aforementioned prophylaxis, the number of CDI cases during the year decreased significantly from 18 to 2, and after its termination, a significant increase in the number of CDI cases to 14 was observed (respectively, 1.0% vs. 0.1% vs. 0.8% hospitalized patients in the nephrology ward in these periods). The absence of severe forms of CDI during prophylaxis with LP299v was also found. The results of this study show that the administration of a probiotic preparation containing *Lactobacillus plantarum 299v* to patients during antibacterial treatment reduces the incidence of *Clostridioides difficile* infection in the nephrology ward [51].

## 10. Treatment of *Clostridioides difficile* Infection

According to the current guidelines of the *Infectious Diseases Society of America* (IDSA) and the *Society for Healthcare Epidemiology of America* (SHEA) published in 2017, the main drugs used in the treatment of CDI are vancomycin and fidaxomicin administered orally [52]. CDI treatment in CKD patients is the same as in the general population.

Depending on the severity, presence of recurrence of CDI there are a number of types of recommended algorithms. For the first mild or severe episode, vancomycin at a dose of 125 mg orally four times a day for 10 days or fidaxomycin at a dose of 200 mg orally twice a day for 10 days is recommended. In the case of unavailability of both above mentioned drugs, it is allowed to use metronidazole in a dose of 500 mg orally, 3 times a day for 10 days. Metronidazole is not recommended as an alternative treatment for severe CDI. In the case of a fulminant first episode of CDI (CDI accompanied by hypotension or shock, intestinal obstruction or toxic megacolon), vancomycin in a dose of 500 mg 4 times a day orally or enteral (via a nasogastric tube) is indicated. In the event of intestinal obstruction, vancomycin is administered in the form of rectal enemas. Moreover, it is recommended to use metronidazole in a dose of 500 mg every 8 h intravenously. The treatment regimen for the first recurrence of CDI depends on the type of treatment used for the first episode. If metronidazole has been used in the past, vancomycin administration is recommended in a dose of 125 mg orally 4 times a day for 10 days. If vancomycin or fidaxomicin were used, treatment of recurrent CDI should be started with prolonged therapy with decreasing and pulsating doses of vancomycin orally. The proposed dosage and duration of such therapy are 125 mg 4 times a day for 10–14 days, 2 times a day for the next week, once a day for another week, and then every 2 or 3 days for the next 2–8 weeks. Fidaxomicin in a dose of 200 mg orally twice a day for 10 days may also be considered instead of vancomycin therapy. In the case of a second or subsequent recurrence, it is recommended to use decreasing and pulsating doses of vancomycin, using vancomycin at a dose of 125 mg orally 4 times a day for 10 days, and then rifaximin 400 mg 3 times a day for 20 days (this drug has no antibacterial effect against *Clostridioides difficile* but accelerates the achievement of eubiosis), the use of fidaxomicin at a dose of 200 mg orally twice a day for 10 days, or fecal microbiota transplantation (FMT) [52].

There were no clear guidelines for treatment regimens in case of simultaneous occurrence of peritonitis and CDI in a group of peritoneal dialysis patients. Shah et al., based on their own experience, propose a choice of treatment depending on the peritonitis etiology. It is advisable to perform dialysis fluid sample collection for microbiological culture and initiate empiric intravenous antibiotic therapy with tigecycline and routine CDI antibiotic therapy (i.e., vancomycin or fidaxomicin orally) at the same time. Tigecycline has antimicrobial activity against Gram-positive organisms, including MRSA and VRE, Gram-negative bacteria and anaerobic bacteria. Tigecycline is also approved for the treatment of intra-abdominal infections, including peritonitis, so it can be an alternative to the routinely used cephalosporin antibiotics in the therapy of peritoneal dialysis-related peritonitis. Tigecycline is not routinely recommended for CDI treatment; however, its efficacy against *Clostridioides difficile* has been confirmed in clinical trials and it can be used as a complementary therapy. In the case that bacterial etiology of peritonitis is other than *Clostridioides difficile,* it is recommended to start antibiotic therapy in accordance with the obtained antibiotic susceptibility test results and to continue the current CDI pharmacotherapy. If *Clostridioides difficile* is cultured in a dialysis fluid sample or the microbiological culture result is negative, it is recommended to maintain the current pharmacotherapy with tigecycline and continue the current CDI pharmacotherapy [40].

Recurrent CDI might be treated with FMT, especially in the case of failure of pharmacological treatment [52,53]. Stool samples are obtained from a healthy person by homogenization and mechanical purification of a liquid suspension in physiological saline. The substance obtained in this way is administered directly into the recipient’s intestines through a nasogastric tube or during the procedure of gastroscopy or colonoscopy. Such treatment is very effective; however, the long-term metabolic effects of intestinal microbiota transplantation are not yet known [54]. In a study by Van Nood et al. comparing the effectiveness of FMT and vancomycin in 81% of patients treated with FMT, no recurrence was observed, compared to 27% of patients without CDI recurrence treated with vancomycin (*p* < 0.001) [55].

There are no published studies on the use of FMT in CKD patients. Friedman-Moraco et al. reported cases of two patients after organ transplantation (kidney and lung transplantation) with recurrent CDI, in whom FMT treatment proved to be safe and effective [56]. In a study by Lin et al. involving five patients after organ transplantation (four patients after kidney transplantation and one patient after pancreatic transplantation) with recurrent CDI, four patients had no CDI recurrence after one FMT. The most common side effects of FMT in these patients were transient abdominal pain and constipation [57].

As was already mentioned, CDI treatment in patients with CKD follows the scheme for the general population. The vancomycin and fidaxomicin used in the treatment of CDI are not absorbed in the gastrointestinal tract, therefore there is no need to modify the dose of these drugs in patients with renal insufficiency.

## 11. *Clostridioides difficile* Infection in the Pediatric Population

As in the adult population, CDI incidence in children has tended to increase since the year 2000. In the population study of children by Khanna et al. covering the years 1991–2009, there was an increase in the incidence of CDI among pediatric patients from 2.6 to 32.6 per 100,000 children [58]. Based on epidemiological data, Wendt et al. showed that 71% of CDI cases identified with a positive stool test for *Clostridioides difficile* in children are community-acquired infections [59]. The CDI risk factors in children are similar to the adult population. An additional risk factor for the pediatric population is the presence of gastrostomy or jejunostomy. Severe and complicated CDI are less common in children than in adults [60,61]. Recommendations for CDI diagnosis are different depending on the children’s age. Due to frequent occurrence of an asymptomatic course of toxigenic *Clostridioides difficile* in infants, testing for CDI should never be routinely recommended for neonates or infants ≤12 months of age with diarrhea. For children aged 1 to 2 years CDI test should be performed after excluding other infectious (viral—mainly rotavirus) or non-infectious causes. In children over two years of age, CDI testing is recommended for prolonged or worsening diarrhea with risk factors such as inflammatory bowel disease, impaired immune system function, contact with healthcare system, or recent antibiotic therapy [52].

In the treatment of CDI in children, metronidazole, vancomycin and rifaximin are used, depending on the form of CDI in pediatric doses. Oral metronidazole or vancomycin is recommended for the first episode of CDI or the first mild to moderate relapse of CDI. Pediatric doses of oral vancomycin are 10 mg/kg/dose 4 times per day for 10 days, maximum dose 125 mg 4 times per day. Pediatric doses of oral metronidazole are 7.5 mg/kg/dose 3 or 4 times per day for 10 days, maximum dose 500 mg 3 or 4 times per day. In case of first episode of severe CDI oral vancomycin is recommended. In this case, metronidazole is recommended only as an additional drug administered by the intravenous route. Pediatric doses of intravenous metronidazole are 10 mg/kg/dose 3 times per day for 10 days, maximum dose 500 mg 3 times per day. For children with a second or subsequent relapse of CDI, it is recommended to use oral vancomycin in the form of decreasing or pulsating doses—10 mg/kg/dose with maximum dose 125 mg 4 times a day for 10–14 days, 2 times a day for a next week, once a day for another week, and then every 2 or 3 days for the next 2–8 weeks. It is also possible to use oral vancomycin for 10 days, then rifaximin orally for 20 days, but there are no pediatric doses of rifaximin—it is not registered for use in children <12 years of age. In the case of repeated recurrences of CDI despite standard CDI treatment, intestinal microbiota transplantation may be considered [52].

There are no separate recommendations for CDI in the pediatric CKD population or in children undergoing renal replacement therapy.

## Figures and Tables

**Table 1 jcm-10-00196-t001:** Epidemiological studies analyzing incidence and characteristic of CDI in chronic CKD patient.

Authors	Study Design	Number of Subjects	Study Period	Results	
				CDI Incidence and Mortality in CKD Patients	Other Outcomes Associated with CDI in CKD Patients
Keddis et al. [6]	Retrospective case-control study Hospitalized patients from general population	162,000,000 patients 8,030,000 with CKD	1 January 2005–31 December 2009	CDI rate in CKD patients was 1.49% compared with 0.70% in patients without CKD (*p* < 0.001) Patients with CKD and CDI had increased in-hospital mortality (aOR, 1.55; 95% CI, 1.52–1.59; all *p* < 0.001) compared to CKD patients without CDI	Long-term dialysis patients with CKD were more than two times as likely to develop CDI than non-CKD patients and 1.33 times more likely than CKD patients not undergoing dialysis (all *p* < 0.001) Patients with CKD and CDI had longer hospitalization time, higher colectomy rate (aOR, 2.30; 95% CI, 2.14–2.47) and more frequent dismissal to a health care facility (aOR, 2.22; 95% CI, 2.19–2.25) compared to CKD patients without CDI
Phatharacharukul et al. [7]	Systematic review and meta-analysis	162,218,041 patients	1998–2014	Pooled RR of CDI in patients with CKD and ESRD were 1.95 (95% CI 1.81–2.10) and 2.63 (95% CI 2.04–3.38), respectively	Pooled RR of recurrent CDI in patients with CKD was 2.61 (95% CI 1.53–4.44)
Thongprayoon et al. [8]	Systematic review and meta-analysis	116,875 patients	1998–2012	Pooled RR of mortality risk of CDI in patients with CKD, ESRD and CKD or ESRD were 1.76 (95% CI: 1.26–2.47), 1.58 (1.37–1.83) and 1.76 (1.32–2.34), respectively	Pooled RR of recurrent CDI in patients with CKD was 2.73 (95% CI: 1.36–5.47) Pooled RR of severe or complicated CDI in CKD patients was 1.51 (95% CI: 1.00–2.28)

CDI—*Clostridioides difficile* infection; CKD—chronic kidney disease; aOR—adjusted odds ratio; CI—confidence interval; RR—relative risk; ESRD—end-stage renal disease.

## Data Availability

Data sharing not applicable.

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
