# Peer review of "Clostridioides difficile Infection in Chronic Kidney Disease—An Overview for Clinicians"

_jcm, 2021, doi:10.3390/jcm10020196_

Round 1
Reviewer 1 Report
This is a comprehensive review of CDI with moderate focus on CDI associated with CKD. Overall, the review reads well and has excellent details.
There are times, especially in the later part of the review that the author appear to repeat themselves and this can be tightened a bit.
The CKD associations and focus are moderate in the review. Especially, the association of CDI with CKD due to presence of comrobidities or its independence is not commented upon. Since the data is not available, authors may want to establish this more clearly in the section.
CDI is a more concerning problem in PD patients as they can present with aseptic peritonitis due to colitis or true peritonitis and can cause problems with antibiotic therapy. A recent article dissected this in more details: https://www.ncbi.nlm.nih.gov/pmc/articles/PMC7055046/ Authors may want to consider either this or other such articles to highlight the special issue of C diff in PD patients.
Recommend significant editing for english.
Author Response
We strongly agree with reviewer remarks. We have made every effort to improve the linguistic quality of our article as well as to minimize the repetition in the text. The data published so far do not analyze the relationship between the risk of CDI and comorbidity in patients with chronic kidney disease or those who were treated with renal replacement therapy, but it seems correct to believe that they significantly increase the risk of infection - which is covered in in our manuscript. As recommended, we have supplemented the revised version of our article with information on concerning patients treated by peritoneal dialysis technique based on the proposed literature. This new paragraph is marked with a red color in the text of our manuscript.
Reviewer 2 Report
This is an excellent review that provides interesting and useful information about CDI in the CKD population. I have a few suggestions for consideration:
1. In sections 1 and 2 of the review, there are numerous grammatical errors and some spelling errors. For example: “genetic analyzes” should be “genetic analysis”. The word “the” inserted where it should not be and omitted from where it should be. These errors seem to be present only in sections 1 and 2 of the manuscript.
2. Table 1: the bullets are distracting and the information should be more clearly organized into additional columns, rather than bulleted points all combined within a single column. For example one column could be incidence or RR of CDI in CKD and another column would be Outcomes associated with CDI (hospitalizations, mortality, etc).
3. Under section 5, “The role of dysbiosis in the CDI pathogenesis”, I would consider creating a separate section or subsection starting at page 5, line 169, titled “Effects of Pre and Probiotics on Intestinal Microbiota” or something similar, since this is a distinct topic.
4. Can you briefly comment on the incidence, associations and treatment of CDI in the infant and pediatric populations? These differ from that of adults and would be useful to include in this review.
Author Response
We completely agree with reviewer remarks. We have made every effort to improve the linguistic quality of the article. We have modified the layout and data arrangement in Table 1 to make it more clear - the table has separate columns: “CD incidence and mortality in CKD patients” and “Other outcomes associated with CDI in CKD patients”. According to the recommendations we have separated the sub-section entitled "Effects of Pre and Probiotics on Intestinal Microbiota". As recommended, we have supplemented in the revised version of our manuscript with information on the prevalence, symptoms, and treatment of CDI in pediatric patients. There is no data available in the current literature concerning the CDI in children with kidney disease or undergoing renal replacement therapy. All above mentioned remarks were added to the new version of our manuscript.